# A Meta-Analysis of the Relationship between Environmental Regulations and Competitiveness and Conditions for Its Realization

**DOI:** 10.3390/ijerph19137968

**Published:** 2022-06-29

**Authors:** Yanli Li, Jiayuan Li, Luyao Gan

**Affiliations:** College of Forestry and Landscape Architecture, South China Agricultural University, Guangzhou 510642, China; luckdoglee2022@163.com (J.L.); 13533275816@139.com (L.G.)

**Keywords:** environmental regulation, competitiveness, relationship, meta-analysis

## Abstract

This study explores the relationship between environmental regulations (ERs) and competitiveness, and the moderating role of the research level, economic development, industry characteristics, and types of measurement in this relationship. To this end, we conducted a meta-analysis of 30 empirical studies. We found that overall, ERs are positively correlated with competitiveness; the industry characteristics have a significant moderating effect on the ER–competitiveness relationship, and ERs more significantly improve the competitiveness of pollution-intensive industries; and the relationship between ERs and competitiveness is universal across research levels, economic development, and types of measurement. This study extends the previous research by supporting the Porter hypothesis and provides a theoretical basis for governments to strengthen the intensity of ERs for pollution-intensive industries and theoretical guidance for enterprises to respond to ERs.

## 1. Introduction

Due to the worldwide problems of energy resources’ exhaustion, environmental degradation, and climate change, environmental protection has become a major global concern. The market fails where environmental protection is concerned because of the existence of environmental externalities [1], and environmental regulations (ERs) should be established to address and alleviate effects of this nature [2]. However, scholars have debated the possible effect of ERs on competitiveness. The relationship between these two constructs has been estimated as positive [3], negative [4,5], U-shaped [6,7], or nonexistent [8]. This disparity suggests that the relationship may depend on the conditions under which ERs exert various effects on competitiveness. For example, Stoever and Weche argued that the impact of ERs on competitiveness depends on factors such as the type of regulations, contextual variables, and estimation methods [9]. The Porter hypothesis also insists that only with an appropriate combination and a certain intensity of ERs—and under appropriate contextual factors—can ERs enhance the green competitiveness of enterprises [3]. In this study, we consider if ERs can improve competitiveness and if the ER–competitiveness relationship is influenced by research level, economic development, industry characteristics, and types of measurement. 

Meta-analysis is an effective method for analyzing the cumulative results of existing studies by quantitatively integrating findings that allow the assessment of the generalization of relationships [10]. Meta-analysis can not only estimate the average effect size between two variables but can also explore the moderating effect of contextual variables on a relationship [11].

To our knowledge, this study conducted the first comprehensive analysis of the ER–competitiveness relationship and the conditions for its realization. Our research contributes to the existing literature as follows: First, we evaluated the generalization of the ER–competitiveness relationship by means of a meta-analysis based on 30 studies. We explored the moderating role of the research level, economic development, industry characteristics, and measurement by a meta-regression analysis of 30 independent empirical studies. This study provides a theoretical basis for governments to strengthen the intensity of ERs for pollution-intensive industries and theoretical guidance for enterprises to respond to ERs. The remainder of the paper is structured as follows: Section 2 summarizes the different relationships between the two constructs and proposes the hypotheses. Section 3 introduces the procedure of meta-analysis. The results are presented and discussed in Section 4. Section 5 concludes the paper.

## 2. Literature Review and Hypotheses

### 2.1. Relationship between Environmental Regulations and Competitiveness

The contention that ERs can enhance competitiveness is controversial. The existing research on the relationship does not go beyond the frameworks of the costly regulation hypothesis [12] and the Porter hypothesis [3]. The focus of the debate is on whether there is a win–win outcome between ERs and competitiveness and on the conditions for the realization of this outcome.

### 2.2. Positive Effect

Porter and Linde argued that ERs can trigger innovation and can fully or partially offset the costs of compliance, leading to absolute competitive advantages. Innovation includes the design of a product or service, the segments it serves, how it is produced, how it is marketed, and how it is supported [3]. Some contend that adherence to environmental laws and regulations has become a win–win strategy for enterprises. For example, Mazzanti and Zoboli evaluated data from 29 sector branches in Italy and found that ERs in Italy increased the labor productivity of enterprises and generated economic benefits [13]. 

Scholars have further developed the theoretical basis of Porter’s hypothesis from the perspectives of organizational behavior, market failure, organizational failure, and production/product structure. Ambec and Barla reformulated the Porter hypothesis from the perspective of agency costs. They pointed out that ERs may reduce agency costs for managers and enhance pollution-reducing innovation while simultaneously increasing firms’ benefits [14]. Lipscomb explored the impact of ERs on multiproduct enterprises’ choice of product portfolios, using a sample of more than 2300 large firms in the Indian manufacturing sector from 1997 to 2005 [15] and found that enterprises reacted to strengthened laws and regulations by producing more clean products in their portfolios [15]. According to the instrumental stakeholder theory, compliance with ERs is beneficial for companies in maintaining better relationships with core stakeholders, obtaining key resources mastered by core stakeholders, and maximizing long-term profits [16]. 

### 2.3. Negative Effect

The traditional view adopts the costly regulation hypothesis as its starting point. Some economists hold that ERs lead to increased abatement costs of pollution and a decreased firm profitability [17]. Lanoie et al. [18], Rubashkina et al. [19], and Deschenes [20] reached consistent conclusions. They found that although ERs impose a positive impact on corporate research and development activities, they have an insignificant effect on corporate total factor productivity. Furthermore, ERs lead to a loss of competitive advantage through an increase in corporate production costs. Gollop and Robert’s study of the SO_2_ emission limit policy in the United States confirmed that ERs are not conducive to increasing the productivity of the power industry [21].

### 2.4. Uncertain Relationship

There are some uncertainties in the process of translating ERs to a competitive advantage. Some studies have found that a simple linear relationship between ERs and competitiveness does not exist. The relationship is either uncertain, positive [22], negative [23], U-shaped [24], inverted U-shaped, or nonexistent [8]. ERs exert a threshold effect on competitiveness, underlining the heterogeneity of influences from different ERs [25]. Almost all possible relationships are supported by empirical evidence. For example, Zheng et al. conducted a study examining 11 provinces and cities in China’s coastal areas and identified a U-shaped correlation between ERs and economic efficiency [24]. Wang et al. analyzed the data of industrial sectors in the Organization for Economic Co-operation and Development (OECD) countries and indicated a threshold effect of ERs on competitiveness [26]. When ERs are lower than the threshold, they promote an increase in green productivity. However, when regulations exceed the threshold, the cost of compliance exceeds the benefits of innovation. Testa et al. conducted a regression analysis of the relationship between the building and construction sector through interviews of managers in three European regions and found that a well-designed direct ER program appears to be the most effective policy instrument for promoting a positive impact on innovation and intangible performance, whereas economic policy tools adversely affect the economic performance of enterprises [27]. 

Based on the costly regulation hypothesis and the Porter hypothesis, we propose the following hypothesis:
**H1.** *ER has an effect on the competitiveness of enterprises, industries, or countries.*


### 2.5. Moderating Effect

The realization of the Porter hypothesis depends on the context. Industry characteristics, regions, time, and enterprise scale may all affect the direction or intensity of the impact of ERs on competitiveness [12,28].

#### 2.5.1. Industry Characteristics

Due to differences in resources, product characteristics, production processes, customer demands, market structure, and other aspects across various industries, there may be disparities in the effect of ERs on competitiveness. Different industries may respond to environmental policies differently [29]. Since pollution-intensive industries have more negative environmental externalities, they become the main targets of ERs [1,30], and ERs can have a greater impact on pollution-intensive industries [29]. Furthermore, there are more changes in production processes, innovations, and product choices in pollution-intensive industries as these industries increase investments in pollution control, clean production technology, and pollution treatment facilities in response to stringent ERs [30]. Compared with companies in so-called clean industries, the costs of compliance with ERs in pollution-intensive industries are significantly higher; correspondingly, competitiveness may also be affected to a greater degree. For example, Wang and Shen asserted that the impact of ER on green productivity is affected by industry characteristics, which have a significant positive impact on clean production industries and a lagging effect on pollution-intensive industries [31]. Therefore, the effect of ERs on competitiveness may be influenced by the pollution-intensive nature of the industry. We propose the following hypothesis:
**H2.** *The pollution-intensive nature of the industry may influence the effect of ERs on competitiveness.*


#### 2.5.2. Economic Development

There are relatively few cross-regional and cross-cultural comparative studies on the effect of ERs on competitiveness. Sen analyzed the effect of the country-level ownership structure on the relationship and found that ERs have different influencing mechanisms in different economies with varying levels of agency problems [32]. Different countries have disparate social systems, resources, public environmental awareness, and other factors that may affect the ER–competitiveness relationship. On the one hand, companies in developed countries function in an environment with relatively rich resources, high public environmental awareness, and strong law enforcement capabilities. A more mature market mechanism and social system enable companies to better obtain the core resources needed to secure competitiveness by following ERs [31]. 

On the other hand, in developing countries, the enforcement of ERs cannot be guaranteed due to a relative lack of resources, a changing economic environment, and a developing social system. Enterprises are motivated to engage in speculative behaviors rather than comply with ERs, and, as Tang et al. noted, ERs “may lead to bad money driving out good” [29]. 

The environment in developing countries is dynamic, complex, and uncertain [29]. Economic uncertainty may affect enterprises, nongovernmental organizations, consumers, and regulators [33]. Customers and the competition faced by enterprises are unpredictable, and market trends and industry innovations change frequently. Enterprises may engage in the trimming of environmental strategy and shift the configurations of investments in the environment through compliance with ERs to adapt to an economic setting of uncertainty [33,34] and, in this environment, may pursue new market opportunities and gain competitiveness. We propose the following competing hypotheses:
**H3a.** *Compared with developing countries, ERs in developed countries have a greater effect on competitiveness.*
**H3b.** *Compared with developed countries, ERs in developing countries have a greater effect on competitiveness.*


#### 2.5.3. Measurement Methods and Research Level

Competitiveness is measured in the literature by a single indicator or a comprehensive index. Single financial performance indicators as measures of enterprise or industry competitiveness include Tobin’s *Q* (i.e., the *Q* ratio), return on assets, and return on equity metrics [35]. Comprehensive indexes involve multiple indicators, including total factor productivity, green total factor productivity, and ecological efficiency, and every indicator is weighted [36,37]. Since different weights are chosen in different studies, heterogeneous results are obtained. 

There is still no widely accepted measurement of ER. Existing measures include single policy events, comprehensive indicators, questionnaires, environmental costs, shadow prices, the environmental performance/output, environmental taxes, the number of environmental policies, environmental investment, and emission limits. These measurements can be divided into three categories [26]—the single indicator, multidimensional indicator, and comprehensive index. Pollution abatement and control expenditures (PACE) are the most accepted single indicators and cover the direct investment and expenditure in pollution treatment and control [19,38,39,40]. Multidimensional indicators are mainly used to distinguish the different impacts of various types of ERs and are estimated by different measurement indicators [41,42]. A comprehensive index considers diverse aspects of ERs and can effectively overcome the limitations of the narrow design of single indicators and multidimensional indicators. The Environmental Policy Stringency (EPS) index, developed by OECD countries, includes market and nonmarket indicators and is a widely used comprehensive index in the literature [26,43]. In addition, research on the relationship under consideration has a broad scope and includes business-, industry-, province/region/city-, and country-level research. 

Due to the lack of uniformity in measurement methods, there may be differences in results [44,45]. Consequently, the testing of the effect of the measurement and the scope of research on the relationship is essential. We propose the following assumptions:
**H4a.** *The measurement of ER affects the ER–competitiveness relationship.*
**H4b.** *The measurement of competitiveness affects the ER–competitiveness relationship.*
**H4c.** *The research-level affects the ER–competitiveness relationship.*


## 3. Method

This study carried out a meta-analysis following Lipsey and Wilson’s research procedure to control for possible deviations and increase the credibility of analysis. Comprehensive Meta-Analysis (CMA) V3 software was used for the meta-analysis [46].

### 3.1. Literature Search

To ensure the inclusion of the studies of appropriate quality, two broad eligibility criteria were used. First, we systematically searched the ScienceDirect, Wiley, SpringerLink, and Web of Science databases for empirical articles published from 1995 to 2020 (as Porter systematically published his hypothesis from 1995). Keywords included Porter hypothesis, ER, competitiveness, and alternatives such as combinations of environmental policy, performance, and others. Next, the references of the review articles and empirical articles were searched to identify any additional relevant studies. This initial search yielded 211 empirical papers, including journals and working papers. 

By combining the research theme and the requirements of the meta-analysis method, we expected the final selected papers to meet the following conditions: Include the ER and competitiveness variables and;Represent an empirical analysis and report the sample size, effect size, or other indicators that could be converted into the effect size.

If there were multiple effect sizes in the study, either the average effect size or one of the effect sizes was used. Studies that did not report correlation coefficients or other indicators that could be converted into correlation coefficients were excluded from the analysis [47]. Finally, 30 articles were included in the final analysis, with a cumulative sample size of 2,333,459.

### 3.2. Effect Size

The correlation coefficient between ERs and competitiveness was used as the effect size. We collected correlation coefficients or other convertible data indicators (i.e., regression coefficients and the *t* value) that could be converted into correlation coefficients [47]. When each correlation coefficient was used as the effect size, it was necessary to use Fisher’s z to convert the coefficients and to use the converted coefficients as the effect size for the analysis [46]. This study included 30 effect sizes.

### 3.3. Coding the Studies

This study followed Lipsey and Wilson [46] for data coding. The data included two parts: effect size and research characteristics. We coded every study of the effect size, sample size, industry characteristics, economic development, measurement of ER, competitiveness, and the research level. The research characteristics included the author, publication year, and journal of publication. On completion of the coding, the accuracy was ensured by two scholars performing a secondary coding for one-third of the studies. The results of the primary and secondary coding were compared, and the consistency coefficients were 95% and 100%, respectively. For the inconsistent coding content, a consensus was reached through the reviews of the original text and discussions.

### 3.4. Homogeneity Analysis

We used the *Q* statistic for the homogeneity analysis of the effect sizes. The *Q* values were compared with a chi-square distribution [46,47]. Samples with heterogeneity can be analyzed by meta-regression analysis. 

If the *Q* statistic is statistically significant, it indicates that these effect sizes represent a heterogeneous distribution and that the random-effects model should be used. If the *Q* statistic is not statistically significant, there are similar results with the fixed-effects model and the random-effects model [44]. I^2^ measures the size and degree of heterogeneity among multiple studies and describes the percentage of the variance between studies caused by non-sampling errors in the total variance. An I^2^ greater than 50% indicates significant heterogeneity.

### 3.5. Publication Bias

Our meta-analysis used the relevant literature as the sample, and it was, therefore, necessary to collect the whole population of relevant studies. However, most journals prefer to publish articles with significant results, which may have prevented some studies showing no significant results from being included in the meta-analysis population, leading to publication bias. 

We used the fail-safe *N* to evaluate publication bias. Scholars generally calculate the fail-safe *N* to assess the severity of publication bias. The fail-safe *N* means *N* unpublished studies need to be added to the samples of the meta-analysis if it is expected to reach a lower final correlation coefficient or significance level than a certain critical value. The larger the gap between *N* and the sample, the smaller the publication bias [45].

## 4. Results and Discussion

Table 1 documents the industry characteristics, economic development, the sample size, the observed correlations for the entire sample, the Z-values, and the weights of the fixed- and random-effects models.

### 4.1. Effect Size Distribution and Publication Bias

Figure 1, the funnel plot of the publication bias test, shows the distribution of the effect size. The funnel chart assumes that the accuracy of the estimated effect size increases with the increase in sample size, and its width gradually narrows with the increase in accuracy. The plot ultimately approaches a point-like shape similar to an inverted symmetrical funnel. Most of the studies are located at the top of the funnel chart and in the vicinity of the average effect size, and only a few appear at the bottom and outside of the funnel chart. This indicates a low possibility of publication bias in this meta-analysis. The study further calculated the fail-safe *N* to evaluate publication bias. The fail-safe *N* corresponding to the variables of ERs and competitiveness is 16,695. That means that 557 unpublished studies are required for each observed study to render the results insignificant because the actual sample size is 30. Therefore, this study is not affected by publication bias.

### 4.2. Heterogeneity Test

The heterogeneity test verifies whether the heterogeneity of multiple studies is statistically significant. The results (*Q* = 7729.649, *p* = 0.000) imply that each effect size is heterogeneous. The random-effects model should be used, and it can simultaneously consider the variation between studies and estimate the average of the effect size distribution. This can prevent the underestimation of the weight of small samples and the overestimation of the weight of large samples and can generate larger confidence intervals, thus leading to a more conservative conclusion [44]. In this study, I^2^ = 99.625% indicates that 99.625% of the variance was caused by true differences between the effect sizes, and only 0.375% of the observed variance was caused by random errors. The value of Tau^2^ is 0.006, which means that 0.6% of the inter-study variation can be used to calculate the weight. 

### 4.3. Main Effects

In the random-effects model, the modified correlation coefficient between ERs and competitiveness was *r* = 0.168 (*p* = 0.000 < 0.001), which is greater than 0.10 [68]; therefore, the correlation is moderate. The 95% confidence interval ranges from *r* = 0.137 to *r* = 0.199. As zero is not included in the confidence interval, it can be concluded that the mean effect size is statistically significant. Therefore, H1 was supported. A one-study-removed analysis (i.e., sensitivity analysis) was conducted to assess the robustness of the analysis. The results showed that the outcome of the meta-analysis has not changed, indicating that the analysis is highly reliable. The mean effect size suggests that only 16.8% of the variation in competitiveness comes from ER.

The results of the random effects analysis showed that ERs are positively correlated with competitiveness. In other words, ERs lead to an increase in competitiveness, an outcome expected by many scholars and policymakers. However, this is inconsistent with the conclusions of Gollop and Roberts [21], Gray and Shadbegian [69], and Lanoie et al. [70]. The difference between the conclusions may be attributed to the following: First, a cumulative result is obtained through meta-analysis and does not exclude the differences in the results of the individual studies. Second, there exists a diversity of the effect mechanism of ERs on competitiveness, and there exist mechanisms to enhance and weaken competitiveness at the same time. The final result depends on which mechanism prevails. Scholars analyze the mechanisms from various theoretical perspectives and obtain different aspects of the relationship.

### 4.4. Moderator Analysis

Due to the heterogeneity between the samples, we conducted a further meta-regression analysis on the potential moderators of the relationship between ERs and competitiveness. There are usually two types of moderators under consideration; one is measurement-concerned, and the other is the research context [71,72]. Potential moderators include the economic development of the sample countries or regions, the research level (i.e., business-, industry-, province/region/city-, and country-level), the independent vs. dependent variable measurement method, and sample industry characteristics (i.e., pollution-intensive or other industries). We coded the above variables in each study and explored their moderating effect on the ER–competitiveness relationship. Table 2 shows the results of the meta-regression analysis.

### 4.5. Industry Characteristics

A meta-regression analysis on the moderating effect of industry characteristics was conducted. With respect to the pollution of the samples in the individual study, the meta-analysis population was classified into two categories—pollution-intensive and other industries. The other studies did not have clear industry characteristics. The results of the meta-regression analysis of the industry characteristics are significant (R^2^ = 0.24, *p* = 0.0000). The correlation coefficient between ERs and competitiveness in the study of pollution-intensive and the other industries are 0.3236 and 0.0938, respectively. Whether the industry is a pollution-intensive one explains the 24% of the inter-study variability. ERs are more significantly improving the competitiveness of pollution-intensive industries. Thus, H2 was supported. 

The industry characteristics of pollution-intensiveness affect the size of the effect of ERs on competitiveness. Accordingly, ERs will more significantly enhance the competitiveness of pollution-intensive industries. This conclusion is inconsistent with the results of Peng et al. [73], who asserted that the market-oriented ERs have a stronger positive effect on the productivity of less pollution-intensive enterprises. Due to high energy consumption and environmental pollution, pollution-intensive industries are currently the main source of damage to the environment and are, therefore, increasingly a primary target of ERs [30]. In pollution-intensive industries, ERs push enterprises to improve energy structure, production process, and the treatment of pollutant emissions, which leads to increased costs. To offset costs, enterprises must innovate to improve efficiency, leading to gains in both environmental protection and competitiveness. In non-pollution-intensive industries, fewer costs are incurred conforming with ERs and, hence, less motivation exists to innovate for efficiency [30]. Hence, environmental policy development should be based on the characteristics of heterogeneous industries to improve policy effectiveness.

### 4.6. Economic Development

Excluding one study that did not specify a country, the remaining 29 studies were divided into two categories based on the different levels of economic development, i.e., developing and developed. Among the 29 studies, 17 had samples from China, accounting for 58.62%. One study had samples from Pakistan, and the rest were from OECD countries. The results of the meta-regression analysis (R^2^ = 0.00, *p* = 0.0000) are shown in Table 2, indicating that the category of economic development does not influence the effect of ERs on competitiveness. Specifically, the relationship between the two variables is not affected by the level of economic development of the countries. H3 was rejected.

### 4.7. Other Moderators

The 30 studies were divided into two categories—subjective and objective measurement methods—to determine whether the measurement of the independent variables affects the ER–competitiveness relationship. The subjective measurement signifies that the measurement of the variables mainly comes from the subjective evaluation of managers or related personnel, while the objective measurement indicates that the measurement of variables mainly comes from objective second-hand data. The same process was performed on the dependent variables. The results of the two meta-regression analyses are insignificant (R^2^ = 0.00, *p* = 0.0000). Thus, the measurement of the variables does not affect the ER–competitiveness relationship. Based on the research scope, the studies were divided into the business, industry, province/region/city, and country levels. Meta-regression analysis was conducted to explore the impact of the research scope. The results show that the scope of research does not affect the ER–competitiveness relationship (R^2^ = 0.00, *p* = 0.0000). Hence, this relationship has a certain universality despite differences in the measurements and research scope. H4 was rejected. 

## 5. Conclusions

This study conducted the first comprehensive analysis of the ER–competitiveness relationship and the conditions for its realization through a meta-analysis of 30 studies. Our conclusions are as follows: On average, there is a moderately positive correlation between ERs and competitiveness, and ERs can lead to increased competitiveness. There is a win–win relationship between the environmental impact and economic development. Our study supports the Porter hypothesis adopted by most scholars;Among the moderators, industry characteristics have a significant moderating effect on the ER–competitiveness relationship. ERs more significantly demonstrate a positive effect on competitiveness in pollution-intensive industries;The economic development, measurement of variables, and research level have no significant impact on the ER–competitiveness relationship. This verifies the generalization of the Porter hypothesis in the above contextual factors.

Our research contributes to the existing literature as follows: First, we support the Porter hypothesis, which is considered controversial. Second, we tested the moderating effect of industry characteristics on the ER–competitiveness relationship. Our findings have important implications for scholars and policymakers. For scholars, future research should focus on the realization condition of the Porter hypothesis instead of the complex relationship between ERs and competitiveness. For policymakers, our results provide a theoretical basis for governments to strengthen the intensity of ERs in pollution-intensive industries through diversified financial, legal, and other related measures [74]. 

This study has limitations, and future scholars can improve on our study in the following respects: The limitations of the meta-analysis place relatively high requirements on the sample studies; the sample cannot realistically include all relevant research. In addition, the moderators of the meta-analysis must be individually coded for each study. This constraint limits the choice of original research. Therefore, it is impossible to explore all potential moderators in the analysis. Other than industry characteristics, there may be other moderators (e.g., culture, ownership) that are worthy of further exploration.The Porter hypothesis is the theoretical premise of most of the previous studies, and the average effect size obtained by our meta-analysis confirms this approach. However, the effect of ERs on competitiveness is a complex mechanism. Therefore, it is more realistic and valuable to identify the multiple effects of ERs on competitiveness and the conditions under which these effects occur instead of focusing on the relationship between the two constructs.Competitiveness is a core variable of concern for governments, industries, and companies. The existing studies mainly used financial performance and productivity indicators to measure competitiveness. However, other forms of competitiveness measures, such as market share, should be used to explore the overall effect of environment regulations.

## Figures and Tables

**Figure 1 ijerph-19-07968-f001:**
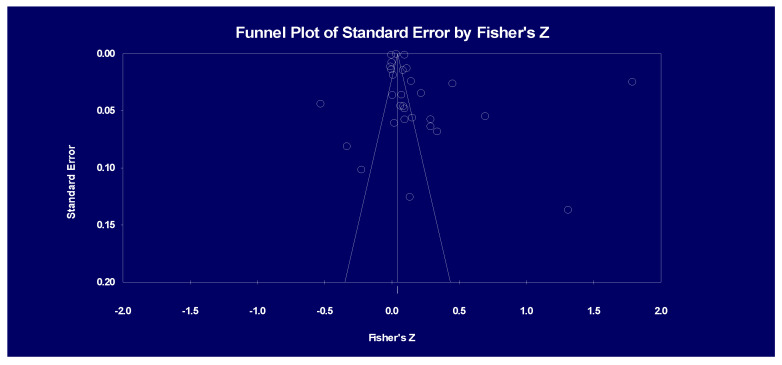
Distribution of effect size.

**Table 1 ijerph-19-07968-t001:** Partial coding information and effect size of meta-analysis.

Author (Year)	Industrial Characteristics	Economic Development	Sample Size	Observed r	Z-Value	Weight of Fixed-Effect Model	Weight of Random-Effect Model
Mi (2018) [48]	None	China	270	0.022	0.36	0.01	2.72
Stöver (2015) [9]	None	Germany	2712	0.013	0.66	0.12	4.16
Zárate-Marco (2013) [49]	None	Spain	153	−0.319	−4.048	0.01	2.09
Rassier (2015) [22]	Polluting	USA	740	0.008	0.217	0.03	3.6
Hu (2017) [39]	None	China	315	0.154	2.745	0.01	2.88
Yuan (2017) [7]	None	China	99	−0.218	−2.174	0.00	1.62
He (2020) [23]	None	China	7208	−0.007	−0.588	0.31	4.32
Javeed (2020) [40]	None	Pakistan	1406	0.426	17.060	0.06	3.95
Yang (2020) [37]	Polluting	China	1569	0.946	70.925	0.07	3.99
Du (2020) [50]	None	China	411,111	0.000	0.000	17.62	4.41
Testa (2011) [27]	None	Three European regions	56	0.865	9.569	0.00	1.07
Costantini (2012) [51]	None	EU	15,453	0.002	0.273	0.66	4.37
Ahmad (2019) [52]	None	China	416,152	0.098	63.097	17.83	4.41
Yuan (2020) [53]	None	China	300	0.284	5.033	0.01	2.83
Tang (2020) [29]	None	China	1,454,899	0.036	43.164	62.35	4.41
Lin (2020) [6]	Polluting	China	464	0.090	1.929	0.02	3.25
Qian (2019) [54]	None	China	330	0.603	12.623	0.01	2.93
Li (2017) [55]	None	China	66	0.137	1.094	0.00	1.21
Nishitani (2016) [56]	None	Japan	5686	0.114	8.615	0.24	4.29
Hwang (2017) [57]	None	OECD countries	4788	0.000	0.086	0.21	4.27
Ghosal (2019) [58]	Polluting	Sweden	245	0.284	4.544	0.01	2.62
Alsaifi (2019) [59]	None	China	752	0.076	2.084	0.03	3.61
Qiu (2020) [60]	Polluting	China	472	0.068	1.466	0.02	3.26
Telle (2007) [61]	Polluting	Norway	427	0.098	2.024	0.02	3.17
Yu (2020) [62]	None	China	299	0.099	1.702	0.01	2.83
Rassier (2011) [63]	Polluting	USA	815	0.218	6.304	0.03	3.66
Hu (2020) [64]	None	China	510	−0.482	−11.828	0.02	3.33
Fu (2020) [65]	None	China	4430	0.085	5.669	0.19	4.26
Peuckert (2014) [66]	None	43 countries	215	0.329	4.970	0.01	2.47
Darnall (2010) [67]	None	Seven OECD countries	1517	0.146	5.981	0.07	4.01

**Table 2 ijerph-19-07968-t002:** Meta-regression analysis results.

Moderators	Tau^2^	I^2^	*p*	R^2^
Economic development	0.0061	99.65%	0.00	0.00
Industry characteristics	0.0046	99.52%	0.00	0.24
Measurement of dependent variable	0.0060	99.64%	0.00	0.00
Research level	0.0060	99.68%	0.00	0.00
Measurement of independent variable	0.0060	99.62%	0.00	0.00

## Data Availability

Not applicable.

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
