# Peer review of "A Meta-Analysis of the Relationship between Environmental Regulations and Competitiveness and Conditions for Its Realization"

_ijerph, 2022, doi:10.3390/ijerph19137968_

Round 1

Reviewer 1 Report

Dear Editors, Authors,

Beginning with the first paragraph of the introduction, I notice several major flaws in the text:

"Economic development generally leads to the overproduction of pollutants in the absence of policy intervention. Pollution is recognized as an outcome of market failure. Given that it represents a public good with negative externality, pollution is difficult to address effectively through market mechanisms [1]."

This is a bold statement attributed to an unknown author.

The authors do not distinguish between public goods, common goods, and negative externalities, nor do they distinguish between economic growth and economic development. Economic development does not result in overproduction of pollution per se; this is an inductive generalization error. Even though it is sometimes criticized, the existence of the Kuznets environmental curve is a testable hypothesis on a case-by-case basis. The word generally is obviously used here to mean that it happens sometimes and not always. It is then not a deductive-nomological model of explanation. It is also not a scientific statement, because, by using the principle of the counterfactual, one could easily reject the conjecture of economic development being the reason for the overpollution since the authors themselves argue that it generally, and not always leads to overpollution.

Overpollution necessitates a quantitative measurement of the allowable amount of pollution in order to determine how much exceeds that limit. This is referred to as a quota. And there could be a market for such quotas, as there is with the EU-ETS. While there is no general market for pollutants, there are markets for specific pollutants, such as the EU-ETS and similar schemes, that show the opposite: where there is a market for pollutants (GHGs), there is a significant decrease in emissions. To claim that markets fail in the case of pollution, one must also claim that no market for pollutants has ever reduced pollution, which is also false. A counterfactual, once again, refutes the conjecture.

Sentences are ambiguous. What, for example, is a public good with negative externalities? Pollution? The environment is not mentioned. The environment, on the other hand, is not a public good. It is a public common good. And there is a distinction between the two. The distinction is known as rivalry in consumption, which public goods do not have.

In addition, I disagree with the notion that meta-analysis is a statistical method. It is more of a review of the cumulative results of other authors' statistical methods of analysis.

Because meta-analyses are compilations, they appeal to readers and are frequently cited, which benefits journals by increasing the impact factor. As a result, I am hesitant to reject this paper outright, preferring to have it resubmitted once it has been improved.

Rather than being proven, inductive-statistical hypotheses are tested and rejected. By rejecting the null, we are not automatically accepting or proving the research hypothesis! For the time being, we will accept it! For more information, please see Karl R. Popper's "The Logic of Scientific Discovery." A scientific statement cannot be proved in an inductive-statistical model of explanation because one case of differing empirical evidence can disprove the entire statement.

This is a meta-analysis, according to the authors. There should be at least 100 sources if this is the case. A true meta-analysis is a challenging task. Consider meta-analyses published in the Journal of Economic Literature, where articles frequently have over 500 references.

I strongly advise the article to be rejected at this time, but I do recommend that it be resubmitted at some point in the future when it is ready. As already noted, true meta-analyses are of interest to readers.

Best regards,

Reviewer

Author Response

I thank you for your thoughtful suggestions and insights. The manuscript has benefited from your insightful suggestions. I look forward to working with you to move this manuscript closer to publication in International Journal of Environmental Research and Public Health. The manuscript has been rechecked, and appropriate changes have been made according to your suggestions. The responses are given below. 1. I have rewritten the background section after receiving your thoughtful analysis. Please see the first paragraph for the modifications. 2. I agree with your opinion that meta-analysis is an analytical method for systematic reviews, and is not a statistical method. I have modified the text accordingly and deleted the word “statistical.” 3. I was persuaded by your thoughtful observation that inductive-statistical hypotheses are tested and rejected. I have modified the text accordingly. 4. As you pointed out, the wider and deeper the literature review, the more reliable the conclusions. However, the number of articles included in the meta-analysis depends on not only the abundance but also the quality of the relevant literature. There are screening criteria for the meta-analysis, and not all relevant articles can be included. Furthermore, meta-analyses do not necessarily contain more than 100 articles, and in fact, meta-analyses with fewer than 100 articles are common. For example, one highly cited meta-analysis contains only 25 articles. In this study, we systematically searched the ScienceDirect, Wiley, SpringerLink, and Web of Science databases for empirical articles published from 1995 to 2020. This initial search yielded 211 empirical papers, which is not rich. Finally, only 30 articles met the predefined requirements for relevance and quality criteria.

Reviewer 2 Report

Dear authors,

I congratulate on the paper. I enjoy reading it and I find it quite interesting for the journal.

I suggest some recommendations.

First, the hipotheses 1a an 1b could be summarised in one. For example, H1, ER have an effect on the competitiveness of enterprises, industries, or countries.

Second, in the conclusion section, the main contributions of the study should be highlighted. 

Third, some minor mistakes have to be corrected, such us line 391 and 510.

Author Response

I thank you for your thoughtful suggestions and insights. The manuscript has benefited from your insightful suggestions. I look forward to working with you to move this manuscript closer to publication in International Journal of Environmental Research and Public Health. The manuscript has been rechecked, and appropriate changes have been made according to your suggestions. The responses are given below. 1. I accepted your advice, and summarized Ha and Hb. 2. I rewrote the conclusion section to better highlight the main contributions of this study. 3. I checked the entire paper and corrected all errors.

Reviewer 3 Report

Recommendations for the authors of the article:

1. The article should expand the "Literature Review" section. Particular emphasis should be placed on linking competitiveness with economic development in terms of investment opportunities in the environment. I propose to take into account the following points: https: // doi. org / 10. 3390 / jrfm14010015, https: // doi. org / 10. 3390 / su12166446, https: // doi. org / 10. 1016 / j. lrp. 2014. 07. 001, https: // doi. org / 10. 3390 / pl15010353
2. The article does not describe the importance of economic uncertainty for competitiveness in the environmental protection market.
3.The most advantageous forms of financing the development of environmental markets in the studied countries should be taken into account.
4. I propose that the conclusions from the study should be presented in points.

Author Response

I thank you for your thoughtful suggestions and insights. The manuscript has benefited from your insightful suggestions. I look forward to working with you to move this manuscript closer to publication in International Journal of Environmental Research and Public Health. I read three of the four papers recommended by you; however, I was unable to locate the fourth paper. I would be grateful if you could provide me additional details, including the title of the paper. The manuscript has been rechecked, and appropriate changes have been made according to your suggestions. The responses are given below. 1. Following your suggestion, I expanded the literature review section. In particular, I emphasized the importance of economic uncertainty and investment opportunities in environmental markets. Please see section 2.2.2. of the manuscript for the modifications. 2. I modified the manuscript to stress that diversified financial measures should be taken to strengthen ERs. 3. As suggested by you, I modified the format of the final section to present my conclusions in bullet points.

Round 2

Reviewer 3 Report

Dear Authors, I think in this version the article is scientifically, methodologically and empirically on a very good level. Congratulations. I wish you scientific and professional success.